# Extensive Investigation on the Effect of Niobium Insertion on the Physical and Biological Properties of 45S5 Bioactive Glass for Dental Implant

**DOI:** 10.3390/ijms24065244

**Published:** 2023-03-09

**Authors:** Imen Hammami, Sílvia Rodrigues Gavinho, Ana Sofia Pádua, Maria do Carmo Lança, João Paulo Borges, Jorge Carvalho Silva, Isabel Sá-Nogueira, Suresh Kumar Jakka, Manuel Pedro Fernandes Graça

**Affiliations:** 1I3N and Physics Department, University of Aveiro, 3810-193 Aveiro, Portugal; 2Physics Department and CENIMAT-I3N, NOVA School of Science and Technology, 2829-516 Caparica, Portugal; 3Materials Science Department and CENIMAT-I3N, NOVA School of Science and Technology, 2829-516 Caparica, Portugal; 4UCIBIO, Applied Molecular Biosciences Unit, Department of Life Sciences, NOVA School of Science and Technology, NOVA University Lisbon, 2819-516 Caparica, Portugal; 5Associate Laboratory i4HB—Institute for Health and Bioeconomy, NOVA School of Science and Technology, NOVA University Lisbon, 2819-516 Caparica, Portugal

**Keywords:** Bioglass^®^, biomaterial, niobium oxide, osseointegration, antibacterial properties, cytotoxicity, electrical properties, bioactivity, bone regeneration

## Abstract

Dental implants have emerged as one of the most consistent and predictable treatments in the oral surgery field. However, the placement of the implant is sometimes associated with bacterial infection leading to its loss. In this work, we intend to solve this problem through the development of a biomaterial for implant coatings based on 45S5 Bioglass^®^ modified with different amounts of niobium pentoxide (Nb_2_O_5_). The structural feature of the glasses, assessed by XRD and FTIR, did not change in spite of Nb_2_O_5_ incorporation. The Raman spectra reveal the Nb_2_O_5_ incorporation related to the appearance of NbO_4_ and NbO_6_ structural units. Since the electrical characteristics of these biomaterials influence their osseointegration ability, AC and DC electrical conductivity were studied by impedance spectroscopy, in the frequency range of 10^2^–10^6^ Hz and temperature range of 200–400 K. The cytotoxicity of glasses was evaluated using the osteosarcoma Saos-2 cells line. The in vitro bioactivity studies and the antibacterial tests against Gram-positive and Gram-negative bacteria revealed that the samples loaded with 2 mol% Nb_2_O_5_ had the highest bioactivity and greatest antibacterial effect. Overall, the results showed that the modified 45S5 bioactive glasses can be used as an antibacterial coating material for implants, with high bioactivity, being also non-cytotoxic to mammalian cells.

## 1. Introduction

The loss of one or more teeth has become a common problem resulting from trauma, bone or dental pathology, cancer, or simply ageing. According to statistics reported by the American Association of Oral and Maxillofacial Surgeons [1], 69% of adults aged between 35 and 44 have lost at least one permanent tooth due to an accident, gum disease, a failed root canal, or tooth decay. In addition, by the age of 74, approximately 26% of those persons have lost all their permanent teeth. Therefore, the demand for dental implants has increased rapidly. The statistics in 2010 revealed more than 300,000 dental implants placed per year [2]. The clinical success of dental implants is related to their ability to ensure rapid osteointegration and prevent the development of peri-implantitis [3]. These conditions are essential to ensure the long-term success of the implant. Peri-implantitis disease is an inflammatory condition that affects the tissues around dental implants. Just like a natural tooth, bacteria can grow at the implant neck and over time irritate gums, which remain in a chronic inflamed state and can lead to bone loss around the implant(s). Initially, the implant(s) remain clinically stable, without any sign of mobility being evidenced. In the absence of effective treatment, bone lysis continues, ultimately leading to implant loss. Therefore, peri-implantitis is considered a secondary implant failure.

To mitigate bacterial contamination, various measures have been used such as careful disinfection and rigorous aseptic surgical protocols [4]. Infection, which can also occur after surgery, is characterized by bacterial colonization and biofilm formation on implant surfaces. The biofilm is considered the primary etiologic reason for the inflammation of peri-implant tissues [5]. Peri-implantitis-associated biofilms are caused by a plethora of microbial species including anaerobes and facultative aerobes, and *Staphylococcus aureus* plays a predominant role in the development of this pathology [6]. To ensure the long-term success of the dental implant, the selection of the biomaterial to be used is a critical factor. Research on the development of new biomaterials or the manipulation of the structure and composition of existing biomaterials has been carried out to improve the properties of biomedical devices [7,8,9].

Typically, biomaterials for medical prostheses are made of metallic materials, mainly stainless steel and titanium alloys [10,11,12]. However, metal prostheses show dramatic failure due to their higher mechanical properties than those of bone tissue, resulting in necrosis and stress shielding of the tissue in contact. It is known that the long-term use of these materials can lead to an excessive release of metal ions which can promote local inflammation, pain, and even clinical failure [13,14,15]. Besides this issue, the formation of the biofilm problem can also lead to implant failure. Thus, an optimal implant should be able to prevent bacterial adhesion and enhance osteointegration [16,17]. To address these challenges, new progress in biomaterial science has led to the development of bioactive materials for implant coatings [4,18,19].

It has been reported that modifying the surface topography of an implant, by depositing a bioactive coating, improves the mechanical interlocking of the bone with the implant and increases the processes of proliferation and adhesion of osteoblasts [20,21,22]. Among the bioactive materials, silicate-based bioglasses containing calcium and phosphorus are mainly used in dental and orthopaedic surgery [23,24]. The first bioactive glasses were discovered in 1969 by L.L. Hench, called 45S5 Bioglass^®^, with the weight composition of 45% SiO_2_, 24.5% Na_2_O, 24.5% CaO, and 6% P_2_O_5_ [25]. This bioactive glass promotes the nucleation of calcium phosphate hydroxyapatite (HA) and carbonate (HAC) during ion exchanges that occur through contact/reaction with physiological fluids [26,27,28]. The formation of HAC on the surface of bioactive glasses promotes osteoinduction, osteoconduction, osseointegration, and angiogenesis [29,30,31]. A further benefit of using bioglass is the ability to alter its composition by incorporating extra ions that can improve its functionality without being toxic [32,33].

Several reports have shown that metallic ions (including magnesium, zinc, copper, silver, etc.) could be employed to generate antibacterial activity [34,35,36,37]. However, the information concerning their long-term effects on human health is limited. Loading these ions into bioactive systems is an effective strategy to control their release over a long period [38,39]. Moreover, bioactive materials can optimize the response of the biological system by interacting with the adjacent tissues, inducing reactions that promote their development and regeneration [40]. The presence of niobium (Nb) species in biomaterials has been reported to improve their bioactive and mechanical properties [41,42]. Some investigations reported that niobium ions promote the differentiation and mineralization of osteogenic cells [41,43]. The insertion of niobium in metallic alloys and ceramic matrices has shown superior corrosion resistance and low cytotoxicity [44,45]. Although niobium-containing bioactive glasses have recently attracted great interest, the effect of their incorporation on the physical and biological response of the 45S5 bioactive glasses is, according to our knowledge, very low.

For this purpose, 45S5 bioactive glasses containing Nb_2_O_5_ were prepared by a melt-quenching technique. X-ray diffraction, IR, and Raman spectroscopies were used to evaluate the influence of Nb_2_O_5_ on the structure of 45S5 Bioglass^®^. The electrical characteristics of the glasses were studied due to their potential to be electrically charged/polarized, therefore optimizing the osseointegration. Cytotoxicity tests using human osteosarcoma Saos-2 cells were used to assess the biocompatibility of the 45S5 bioactive glasses modified by the Nb insertion. The in vitro investigation of bioactivity was conducted by immersing samples (pellets made of powdered bioactive glass) in simulated body fluid (SBF), which is a solution with the same ionic strength and mineral content as the blood plasma. The changes in bioglasses, in terms of dissolution and precipitation of the protective layer containing calcium phosphates, were evaluated using scanning electron microscopy (SEM) with energy-dispersive X-ray analysis (EDX). The antibacterial activity of the bioglasses was evaluated from their ability to inhibit the growth of the *S. aureus* bacteria, which is commonly associated with implant infections, as well as other bacteria involved in the formation of the pathogenic biofilm.

## 2. Results

### 2.1. Structural Characterization

The XRD patterns, illustrated in Figure 1, consist of typical broad bands, without traces of crystalline phases, which confirm the amorphous character of the glasses. These results match well with the ones reported in the literature for 45S5 glasses [3,46]. The insertion of niobium, up to 8% mole of Nb_2_O_5_, did not promote the formation of any crystalline phase but enlarged the band at smaller angles. This phenomenon was already observed in phosphate glasses with niobium [47].

Figure 2 depicts the features revealed by the FTIR spectra of the glasses. The Si-O-Si stretching modes are ascribed to the bands detected at around 1010 cm^−1^ and 721 cm^−1^ [46,48,49,50,51,52,53,54]. The presence of the non-bridging oxygen ions is shown by the appearance of a band at 912 cm^−1^ that is attributed to the Si-O_NBO_ stretching mode [46,51,53,54,55]. The presence of a shoulder at 596 cm^−1^ is attributed to the bending vibration of the P-O molecule [46,50,53,54,55,56]. The band that appeared at around 497 cm^−1^ is associated with the Si-O-Si bending mode [46,48,49,50,51,52,53,54].

Figure 3 shows the Raman spectra of the bioactive glasses prepared. For a more detailed analysis, the Raman spectra of BGNb1, BGNb2, BGNb4, and BGNb8 were deconvolved by a Gaussian fitting (Figure 4).

The whole range of Raman spectra can be separated into two regions: low wavenumber (500–900 cm^−1^) and high wavenumber (>900 cm^−1^). In the low-frequency region, a broad band at about 630 cm^−1^ is observed for BGNb0 which can be associated with the rocking motion of bridging oxygen in structural units that contain non-bridging oxygen ions (NBO) [57,58,59]. With the insertion of Nb_2_O_5_ in the glass network, it is noted that the band is shifted towards a higher frequency. By fitting and deconvoluting this peak (Figure 4), two bands can be differentiated, one located at 630–680 cm^−1^ and the other at 710–750 cm^−1^, and attributed to the vibrational mode of the Si-O-Si bond and the vibration of Nb-O, respectively, in NbO_6_ octahedra, with a low degree of distortion [60,61,62]. Moreover, an additional broad peak at 780–890 cm^−1^ is observed for the modified bioglass. The deconvoluted Raman spectra (Figure 4), show that this peak could be distinguished as three different bands located at 782–804 cm^−1^, 806–829 cm^−1^, and 857–886 cm^−1^, which are assigned to vibrations of NbO_6_ octahedrons with different degrees of distortion, to the vibration of NbO_4_ units, and to the symmetric stretching of Q_0_ units in the silicate network (SiO_4_^4−^), respectively [63,64,65]. The vibrational bands of NbO_6_ units increased with increasing Nb_2_O_5_ concentration up to 2 mol%, whereas the NbO_4_ vibrational band decreased. A shoulder is observed for the BGNb0 sample at around 857 cm^−1^, corresponding to symmetric stretching of Q_0_ Si units [54,65]. At the high-frequency region, a vibration band can be observed at around 945 cm^−1^ and a shoulder at 1050 cm^−1^. From the result obtained by a deconvolution process (Figure 4), five vibrational modes located at 900–923 cm^−1^, 938–944 cm^−1^, 967–975 cm^−1^, 997–1023 cm^−1^, and 1051–1078 cm^−1^ can be distinguished, which correspond to symmetric stretching of Q_1_ Si, Q_2_ Si, Q_0_ P, and Q_1_ P units, and asymmetric stretching of bridging oxygen in all Q species, respectively [54,57,65].

The sum of the area of Raman vibration bands associated with non-bridging oxygen ions is plotted in Figure 5 as a function of Nb_2_O_5_ concentration. An increase in the number of NBOs for the bioglass with Nb_2_O_5_ content up to 2 mol% can be observed, indicating the existence of an optimal amount of Nb_2_O_5_ that can be added to the base bioglass.

### 2.2. Thermal Analysis

The differential thermal analysis (DTA) spectra of BGNb2 and BGNb8 are shown in Figure 6. The thermograms of both samples demonstrate the presence of a glass transition temperature, T_g_, followed by an exothermic peak, T_c_, ascribed to structural alteration associated with the formation of crystalline phases, and an endothermic peak, T_m_, attributed to the melting point of bioglass. The T_g_ values were determined from the thermograms using the tangent method on the measured DTA curve, above and below the glass transition region. The value presented in Table 1 is the mean value of the abscissas of the tangents’ interceptions. The uncertainty in these measurements is 2%. In a prior study, a comparable thermal response was observed for the 45S5 bioglass [3]. The critical temperatures of the bioglasses modified by Nb_2_O_5_ are reported in Table 1.

### 2.3. Electrical Characterization

Figure 7 depicts the DC conductivity versus 1000/T on a logarithmic scale. It is visible that the conductivity rises with the increase in the temperature, which should be related to the increased mobility of the charge carriers. For temperatures above 300 K, approximately, this variation becomes linear, indicating that the Arrhenius formalism may be used to evaluate the activation energy associated with this thermally activated process. The calculated activation energy at the high-temperature region (inset in Figure 7) for all samples is registered in Table 2. The activation energy (E_a_ (DC)) decreases with the increase in Nb_2_O_5_ concentration, reaching a minimum value at the value of 2 mol%. After, a reversing trend is observed. This behavior is similar to the one observed for the NBO ions amount (Figure 5). On the contrary, an opposite trend was observed for the DC conductivity (σ_DC_), which shows a maximum value for the sample with 2 mol% of Nb_2_O_5_.

The dielectric properties of the samples were analyzed using the modulus formalism, (M* = 1/ε*), to minimize the effect of electrode polarization and conductivity. Figure 8 shows the presence of one dielectric relaxation which is shifted to higher frequencies with increasing the temperature. It should be noted that other formalisms, such as permittivity or impedance, did not exhibit this dielectric relaxation behavior. Therefore, the observed relaxation behavior should be associated with an intrinsic characteristic related to dipole formation between the network modifier ions and the non-bridging oxygen ions.

The AC conductivity (Figure 9) showed the same feature, in resonance with DC conductivity, where the variation increases linearly in the high-temperature region. The results obtained for AC conductivity are registered in Table 2 and show an agreement with DC conductivity results. The bioglass with 2 mol% Nb_2_O_5_ content exhibits the highest AC conductivity. The activation energy value decreases with increasing frequency for all bioglass samples. This can be explained due to the increase in the applied frequency which enhances the electronic jumps between the localized states [66,67].

### 2.4. In Vitro Evaluation of the Biocompatibility

The human osteosarcoma Saos-2 cells were used to evaluate the cytotoxicity of the bioactive glasses prepared. Two sets of samples were tested, which we call the non-passivated and the passivated ones. The difference between the two sets is that the non-passivated is the filtered solution of the pristine bioglass powders that were in contact with the McCoy 5A medium for 24 h. The passivated solution was achieved by changing the medium after 24 h to a new medium, which was kept in contact with the powders for a new 24 h period. The filtered solution was then called the passivated sample. Figure 10 displays the results obtained for the passivated and non-passivated samples. All samples showed an improvement in cell viability when the concentration decreased from 100 mg/mL to 6.75 mg/ mL, as expected. Moreover, it can be seen that the passivated samples exhibit higher cell viability compared to the non-passivated samples. The passivated samples, which better simulate physiological conditions [68], revealed cell viability higher than 80%, except for the BGNb0 and BGNb1 samples that showed toxicity at the 100 mg/mL concentration and BGNb0 at 50 mg/mL, respectively.

### 2.5. Antibacterial Activity

Figure 11 depicts the antibacterial activity of the bioglasses modified by Nb_2_O_5_ insertion evaluated by the observed inhibition halo against the Gram-positive *Staphylococcus aureus* and *Streptococcus mutans* and the Gram-negative *Escherichia coli* bacteria. By the rise of the inhibition halo, we can conclude that the samples have antibacterial activity. An increase in the inhibition halo with increasing the content of Nb_2_O_5_ up to 2 mol% is noted. A decrease in the antibacterial effect was observed with increasing the Nb_2_O_5_ from 2 to 8 mol%.

### 2.6. In Vitro Degradation and Bioactivity Assay

The in vitro bioactivity, i.e., the ability to induce an apatite-like layer formation, of the bioglasses prepared, was investigated using an immersion test in simulated body fluid (SBF). Figure 12 shows the SEM micrographs and the EDS data of the surface of bioglass pellets after immersion in SBF.

The SEM micrographs confirmed the bioactivity of the glasses. The surface morphologies of pellets show the presence of spherical (cauliflower-like) particles characteristic of an apatitic layer formation. With the increase in the immersion time in SBF, this layer becomes thicker. It can be seen that for the samples with an immersion time of 28 days, the size of the particles decreases with increasing the percentage of niobium beyond 2 mol%, and in the case of the sample with the highest load percentage, BGNb8, only small amounts of apatite particles start to appear. The EDS graphs (Figure 12(d1–d5)) confirm the change in the amount of the chemical elements with the soaking time in SBF. The amount of Na and Si atoms tends to decrease with the immersion time, while the amount of P and Ca tends to increase.

The variation of Si and Na atomic percentage and the Ca/P ratio, determined using the data from EDS analysis on the surface of the bioglass samples before and after immersion in SBF solution, are illustrated in Figure 13. An abrupt drop in the atomic percentage of the elements is evident during the initial 2 days of immersion in all samples while no significant change is observed afterward. The samples with lower percentages of niobium, BGNb1, and BGNb2, show a faster decrease in Na and Si percentage in the first 2 d compared to the other glasses, and the value of the Ca/P ratio decreases reaching a value of 1.60–1.80, comparable to that of hydroxyapatite (Ca/P = 1.67) [69,70,71]. The decrease in the atomic percentage of Na and Si in the sample modified with 8 mol% of Nb_2_O_5_ is not significant, in contrast with the other samples, and the Ca/P ratio is considered high.

Figure 14 depicts the variation of pH of the bioactive glass samples after soaking in SBF for different periods, from 12 h up to 28 days. For all samples, the pH initially increases compared to the initial pH of the SBF solution, 7.4, and then, after 2 days of sample immersion, it starts to decrease.

## 3. Discussion

The results obtained by XRD and FTIR do not reciprocate any type of modification in the structure of the glass matrix with the insertion of Nb_2_O_5_. However, Raman spectra clearly showed the appearance of NbO_4_ and NbO_6_ structural groups, located at 800–810 cm^−1^ and 827–862 cm^−1^, for the glasses modified with Nb_2_O_5_ (Figure 3 and Figure 4). As the concentration of Nb_2_O_5_ is raised up to 2 mol%, the quantity of NbO_6_ units, related to niobium ions acting as modifiers in the glass matrix, increases progressively. Exceeding the 2 mol% of Nb_2_O_5_, there is a fractional conversion of NbO_6_ units into NbO_4_ units, which contributes to the formation of the glass network. Similar behavior was already found in other glasses containing niobium [72,73,74]. The presence of NbO_6_ units is predicted to act as a network modifier through the depolymerization of Si-O-Si bonds. These octahedrons can form a chain structure with different degrees of distortion by sharing their vertices with at least two silicon octahedrons. With the increasing NbO_6_ concentration, a higher concentration of NBO is expected [59,72]. The BGNb2 sample exhibits a high concentration of NbO_6_ structural groups (Figure 4), and thus a high concentration of NBO as justified in Figure 5.

The presence of NbO_6_ and NbO_4_ plays a critical role in the thermal response of the bioglass. From Figure 7 and Table 1, it can be observed that the Tg increases with the insertion of a high concentration of niobium, which indicates that an increase in network connectivity occurs due to the presence of a high concentration of NbO_4_, the network former, leading to a decrease in the amount of NBO ions. For the BGNb2 sample, a slight decrease in the characteristic temperatures is observed compared to the bioglass base, which is attributed to the increase in the amount of NBO ions and the weakness in the glass network, facilitating the ion mobility. Similar behaviors were reported in the literature for niobium glass systems [75,76].

The existence of NBO ions could affect the electrical properties of the prepared glasses. It is known that in the bioactive glass system, the conductivity is mainly related to the energy transported by Na^+^ and Ca^+^ ions moving through the glass network [3,77]. For the glasses modified with Nb_2_O_5,_ the presence of NbO_6_ units leads to the degradation of the bioglass network by the formation of more NBO ions. Such structural alterations may have a significant impact on Na^+^ and Ca^+^ mobility. The mobility of such network modifier ions increases with the rise in NBO ions. One can notice in Table 2 that the sample with 2 mol% content of Nb_2_O_5_ exhibits the highest DC conductivity and therefore the lowest activation energy since the decrease in activation energy suggests an increase in the charge carrier’s mobility. The same happens to the AC conductivity, the increase in the conductivity with samples loaded with Nb_2_O_5_ up to 2 mol% should be essentially related to the increase in Na^+^ ions’ mobility.

From the biomedical applications point of view, it is critical to evaluate the cytotoxicity of the bioactive glasses given their intended use as implant coating materials. The evaluation of the bioactive glasses’ cytotoxicity against the human osteosarcoma Saos-2 cells, represented in Figure 10, shows low cell viability at high extract concentration, especially in the case of the samples with no or low content of Nb_2_O_5_. It was found that increasing the content of Nb_2_O_5_ inserted into the bioactive glass network can increase the biocompatibility of materials, which is consistent with the results of previous studies [75,78,79]. The non-passivated samples show lower cell viability compared to the passivated samples, which could be explained by the high rate of ion-exchange reactions that take place during the first 24 h of material–cell culture medium interactions leading to an increase in local pH [52]. Based on the results of passivated samples, which better mimic physiological circumstances, all modified bioactive glasses by niobium insertion can be used in a biomedical application where the extracellular fluid is exposed to bioactive glasses in circumstances corresponding to the 100 mg/mL extract production. The exceptions are the glasses with Nb_2_O_5_ content lower than 2 mol%, which show cell viability less than 80% at that concentration.

An important factor in reducing the risk of implant infection is preventing bacterial colonization. The 45S5 glass particles promote a considerable antibacterial effect against certain oral bacteria mainly due to the change in pH and the osmotic pressure effect [80,81]. The increase in pH to a more alkaline range creates an unfriendly environment for bacteria, resulting in morphological alterations. Additionally, variations in ion concentration in the bacterial environment result in a reduction in pressure across the bacterial cell membrane, which causes bacteria to shrink and therefore damages the cell membrane [81,82]. The antibacterial test results obtained using the agar diffusion method demonstrate the antibacterial properties against Gram-positive and Gram-negative bacteria (Figure 11). The inhibition halo tends to increase with increasing the Nb_2_O_5_ concentration up to 2mol%, which can be due to the high ionic strength of Nb^5+^ ions preventing bacteria from growing by creating a hyperosmotic environment [72,74]. Additionally, the presence of a disintegrated glass network due to the high concentration of NbO_6_ units in those glasses offers a suitable environment for the leaching of alkali metals ions, Na^+^, and Ca^2+^, which could lead to an increase in pH and thus promote the death of bacteria. Beyond 2 mol% of Nb_2_O_5_ insertion, the antibacterial activity of the samples decreased, which could be related to the conversion of certain NbO_6_ structural units into NbO_4_ units, as proved by the Raman analysis.

To assess the in vitro bioactivity of the synthesized bioactive glasses in a biological medium, the SBF immersion test was chosen. This method serves a better understanding of the physicochemical reactions taking place on bioglass in physiological fluids. When evaluating bioglass in vitro, it is critical to take into account changes in the surface chemistry, mainly the formation of an apatitic layer, since it has a significant impact on osteoblast cell adhesion and proliferation [83]. The formation of an apatitic layer is purely physicochemical and does not involve cells. This was observed in the SEM images of bioglasses after the immersion in SBF which is a cell-free solution (Figure 12). The apatite particles are formed on the surface of the bioglass, and with increasing immersing time the particles tend to aggregate, and a dense layer is formed. However, the size of the formed apatite seems to be smaller for the samples with the highest concentration of Nb_2_O_5_ (above 2 mol%) compared to the other samples. To better understand the effect of niobium on the formation of the hydroxyapatite (HA) layer on the bioglass surface, we refer to the mechanism proposed by L. Hench for the original Bioglass^®^ which is adaptable to the majority of bioactive materials [25]. Similarly to the 45S5 bioactive glass, the SiO_4_ tetrahedron serves as the main structural unit of the bioactive glasses containing niobium, which may form bonds with other SiO_4_ tetrahedrons through Si-O-Si bonding, also known as bridging oxygen (BO) ions. Network modifiers alter the structure of the glass by converting BO into NBO, resulting in a drop in the glass network connectivity and thus a rise in the dissolution rate and consequently the release of ions. In fact, when the glass comes into contact with SBF, the alkali and alkaline earth ions (Na^+^ and Ca^2+^) present on the surface of the glass are exchanged with the H^+^ and H_3_O^+^ ions of the medium. This leads to an increase in the pH of the medium as the H^+^ ions are replaced by cations. The rise in pH values promotes the breaking, in the glass network, of the Si-O-Si bonds leading to a faster dissolution of the glass and the formation of silanol units (Si(OH)_4_). These silanol units then condense on the surface of the glass, forming a hydrated silica layer which promotes the nucleation of the carbonated hydroxyapatite. From the results obtained (Figure 13b), a faster drop can be seen in the atomic percentage of Na on the bioglass surface in the first days of SBF immersion. The presence of niobium acting as a network modifier, at high concentrations in the samples with a low percentage of Nb_2_O_5_ (≤2 mol%), creates a higher number of NBO ions and therefore promotes a faster release of the Na^+^ ions. For the samples with Nb_2_O_5_ > 2 mol%, the amount of NbO_4_ units increases, reducing the amount of NBO ions, and therefore the mobility or release of the network modifiers, namely the Na ions, becomes more difficult leading to a reduction in the glass dissolution rate. The same trend is observed for Si, which exhibits faster release from the bioglass with Nb_2_O_5_ concentration below 4 mol%, as indicated in Figure 13a by the faster decrease in the atomic percentage of this element. This proves the enhanced surface reactivity for the bioglasses with niobium load up to 2 mol%. The SEM micrographs in Figure 12 show that after 24 h of SBF immersion, the amount and the size of spheroidal apatite particles are more pronounced in the sample BGNb2, suggesting the existence of an optimal composition for the apatite formation. With longer soaking times in SBF, the concentration of the ions stabilizes and the Ca/P ratio decreases, reaching a value between 1.6 and 1.8 for the sample with niobium content up to 2 mol%, close to that of the HA of the human bones [69,70]. For the samples with higher niobium amount, the value of the Ca/P ratio is above 2, higher than the ideal value of 1.67. This higher value of Ca/P suggests that those samples are less bioactive [84,85], meaning that the apatite structure does not develop easily on the glass surface when exposed to SBF. After 28 days of soaking in SBF, the apatite particles grow and become denser, which indicates the formation of a dense layer of crystalline HA on the bioglass surface, while the growth was reduced for the samples with niobium content of 4 mol% and only the beginning of the formation of small apatite particles was observed for the sample with 8 mol% of niobium loads.

The variation in the pH is indicated in Figure 14. The increase in the pH in the first 48 h was due to the high release of the alkaline metal ions from the bioglass surface, and then it decreases due to the formation of silanol. The incorporation of 2 mol % of niobium into the bioglass resulted in a higher value of pH, which can be explained by the highest degradation rate of this sample and this is consistent with the condition of the creation of an HA-like layer on the surface of the bioactive glass with higher crystallinity [59,72].

## 4. Materials and Methods

### 4.1. Bioglass Preparation

A series of 45S5 bioactive glasses (45% SiO_2_, 24.5% Na_2_O, 24.5% CaO, and 6% P_2_O_5_ (wt%)) modified by the insertion of different amounts of Nb_2_O_5_, from 0 to 8 mol% (designed by BGNb0, BGNb1…BGNb8) was prepared using the melt-quenching technique. In brief, high-purity grade SiO_2_, P_2_O_5_, CaCO_3_, Na_2_CO_3_, and Nb_2_O_5_ (>99.99%) were mixed and ground in a planetary ball mill at 300 rpm for 1 h to homogenize, before undergoing calcination at 800 °C for 8 h. The powder was then placed in platinum crucibles and melted at 1300 °C for 1 h. The melt was quenched between the casting plates to obtain bulk glass samples.

### 4.2. Structural Characterization

Malvern Panalytical Aeris powder diffractometer (CuK_α_ radiation, λ = 1.54056 Å) was used to collect the X-ray diffraction (XRD) patterns at room temperature. The acquisition was performed using a scan step of 0.02° in 1 s in a 2θ angle range of 10–70°.

The FTIR spectra were performed on FT Perkin-Elmer Spectrum BX Spectrometer in the ATR crystal (Golden Gate Diamond ATR Accessory), in the range of 400 and 1300 cm^−1^. The measurements were obtained from bioglass powder dispersed in KBr pellets. During acquisition, the room temperature and humidity were kept at approximately 23 °C and 35%, respectively.

Raman spectra were recorded at room temperature using a Horiba Jobin Yvon HR 800 spectrometer with an Ar^+^ laser (λ = 532 nm). Spectra were collected in a back-scattering geometry between 200 and 1500 cm^−1^ with a 50x lens focused on the sample.

### 4.3. Thermal Analysis

The thermal properties of the glasses were investigated using a simultaneous differential thermal analysis (DTA)/thermogravimetric (TG) measurement. A Hitachi STA 7300 system was employed, under Nitrogen N50 (99.999%) flowing at 200 mL/min and heating at 10 °C/min.

### 4.4. Electrical Characterization

Bulk glass samples were used for the electrical measurements. The samples were polished until parallel surfaces with a thickness of about 1 mm were obtained and then painted with silver conductive paste to form the electrodes. The direct current conductivity (σ_DC_) of the samples was measured with a Keithley 617 electrometer, capable of measuring currents down to 10^−14^ A. This measurement was performed in a temperature range between 200 and 400 K where a voltage of 100 V was applied across the bulk glass. The AC electrical conductivity (σ_AC_) and impedance measurements were also performed in the temperature range of 200 to 400 K, using a precision impedance meter, Agilent 4294, operating in a broad frequency window from 100 Hz to 1 MHz and in the C_p_–R_p_ configuration. The temperature of the samples was regulated by an Oxford Research IT-C4 and monitored using a platinum sensor in both DC and AC measurements.

The complex electric permittivity ε* was calculated with the following equation [86,87]:ε* = ε’ − j ε’’ = C_p_ (d/ε_0_ A) − j d (ω R_p_ ε_0_ A)(1)

At the high-temperature range, the activation energy (E_a_) of the DC and AC conductivities was determined by fitting the data to the Arrhenius model [88,89,90,91].
σ = σ_0_ exp(−E_A_/(k_B_ T)),(2)
where σ_0_ is a pre-exponential factor, E_A_ is the activation energy, K_B_ is the Boltzmann constant, and T the temperature.

### 4.5. In Vitro Evaluation of the Biocompatibility

In accordance with International Standard “ISO 10993-5 Biological evaluation of medical devices—Part 5: Tests for in vitro cytotoxicity” [92], the cytotoxicity of the bioglasses was assessed at various concentrations against the human osteosarcoma cell line (Saos-2 cells, ATCC^®^ HTB-85™). Extracts were produced by placing the bioglass powder in contact with a culture medium, McCoy 5A medium (Merck KGaA, Darmstadt, Germany) at a concentration of 100 mg/mL. For non-passivated extract, the medium in contact with the powder was kept in an incubator for 24 h at 37 °C then filtered with a 0.22 µm millipore filter and stored at 37 °C. For the passivated extract, a new McCoy 5A medium was added to the same bioglass powder and then placed in the incubator for another 24 h at 37 °C. The 96-well plates were seeded with a Saos-2 cell line at a density of 30,000 cells/cm^2^ and placed in an incubator with 5% CO_2_ atmosphere for 24 h at 37 °C. After, the culture medium was removed and on the same plate, negative controls (C-) (viable cells), positive controls (C+) (in which the toxic compound dimethyl sulphoxide was added), non-passivated and passivated extracts with appropriate dilutions (50 mg/mL, 25 mg/mL, 12.5 mg/mL, and 6.75 mg/mL) were placed in an incubator for 48 h. The resazurin cell viability indicator was used to assess cell populations [93]. Using a Biotek ELX800 microplate reader, the optical absorbances of each well were measured at 570 and 600 nm. To confirm the assay’s reproducibility, two biological replicates, with six statistical replicates each, were carried out in this test for each sample.

### 4.6. Antibacterial Activity

The bacterial strains *Escherichia coli* K12 DSM498, *Staphylococcus aureus* COL MRSA (methicillin-resistant strain), and *Streptococcus mutans* DSM20523, were used as models to assess the antibacterial activity of the glasses, as previously described [46]. The method of agar diffusion assay plates, using the two-layer bioassay, was performed with TSB medium with the molten seeded overlay containing approximately 10^8^ CFU/mL of the appropriate indicator bacteria. Bioglass pellets with a diameter of 6 mm were placed in the center of the plates, left for 4 h at room temperature, and then incubated for 24 h at 37 °C. In the case of *S. mutans* the plates were placed in a 5% CO_2_ incubator.

Images of the pellets were taken, and the diameter of the inhibition halo was measured with ImageJ software; each pellet was measured 30 times in several orientations [94]. The results of the eight independent assays for each bacterium were statistically analyzed with an unpaired *t*-test, comparing the bioactive glass base composition with each of the different samples using GraphPad Prism 8.0 software.

### 4.7. In Vitro Bioactivity Assay

The bioactivity evaluation of the bioglasses was conducted by immersing samples (pellets made of powdered bioglass) in simulated body fluid SBF, according to ISO 23317:2017 Standards. The samples were placed in different flasks, soaked in SBF, and remained inside an incubator at 37 °C with continuous oscillating support for 12, 24, 48, 96 h, 14, and 28 days. The SBF solutions were refreshed every 48 h to mimic the biological environment.

To calculate the volume of SBF used for each sample, we used the following formula:V_s_ = 100 mm × S_a_(3)
where V_s_ is the volume of SBF in mm^3^, and S_a_ is the surface area of the pellet in mm^2^.

After soaking, the pellets were collected and gently rinsed with deionized water and then dried at room temperature.

This assay aimed to determine the change in ion concentration and the formation of an apatite-like layer on the bioglass surface over 28 days with the presence of a different concentration of Nb_2_O_5_. For that, TESCAN Vega 3 scanning electron microscopy (SEM) (TESCAN ORSAY HOLDING, a.s., Brno-Kohoutovice, Czech Republic) equipped with energy-dispersive X-ray spectroscopy (EDS) (Bruker EDS) was performed on the glass surface to determine the morphological and compositional changes resulting from the reaction in SBF.

## 5. Conclusions

Bioactive 45S5 glasses modified by the insertion of different amounts of niobium pentoxide, Nb_2_O_5,_ were successfully synthesized using the melt-quenching technique. XRD and FTIR results show that there was no alteration in the glass matrix with the addition of Nb. The characterization using Raman spectroscopy showed the appearance of additional bands for the bioactive glass containing niobium attributed to the distortion of NbO_6_ units and the vibration of NbO_4_. The fractional conversion of the network modifier units of NbO_6_ into the NbO_4_ network former affects the electrical properties of the sample and causes a decrease in the bioactivity and antibacterial effect. The sample with 2 mol% of Nb_2_O_5_ content presented the highest percentage of NbO_6_ units and showed a higher dissolution rate and maximal growth of the HA layer on its surface in the in vitro immersion tests in SBF. Moreover, the evaluation of the antibacterial activity against *E. coli*, *S. aureus,* and *S. mutans* revealed that glass loaded with 2 mol% of Nb_2_O_5_ had the greatest antibacterial effect. We can conclude that the 45S5 bioactive glass modified by the insertion of 2 mol% of Nb_2_O_5_ is more suitable for biomedical applications and can be employed as a coating material for a dental implant without being harmful to osteoblasts cells.

## Figures and Tables

**Figure 1 ijms-24-05244-f001:**
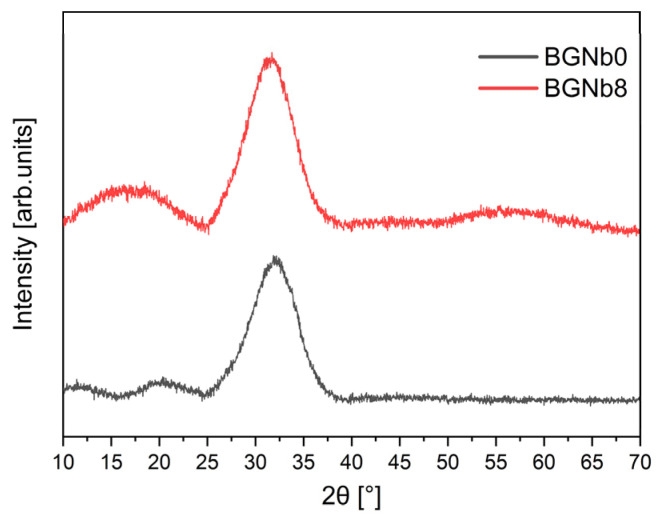
XRD patterns of 45S5 bioglass samples modified by the insertion of 0 and 8 mol% of Nb_2_O_5_.

**Figure 2 ijms-24-05244-f002:**
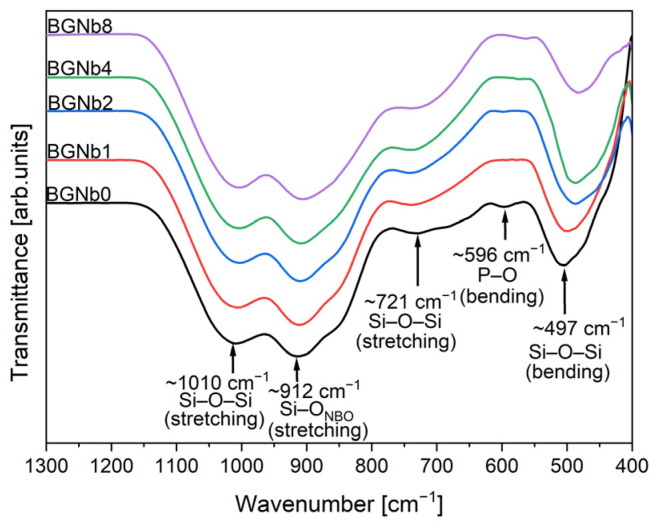
FTIR spectra of bioactive glasses modified by Nb_2_O_5_ insertion.

**Figure 3 ijms-24-05244-f003:**
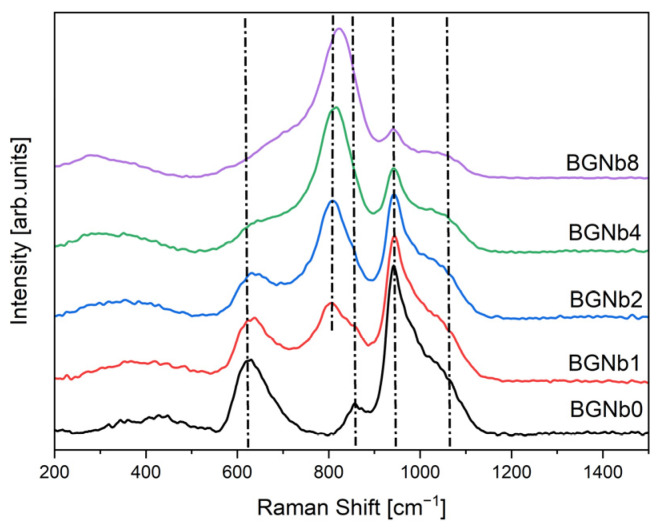
Raman spectra of bioactive glasses modified by Nb_2_O_5_ insertion.

**Figure 4 ijms-24-05244-f004:**
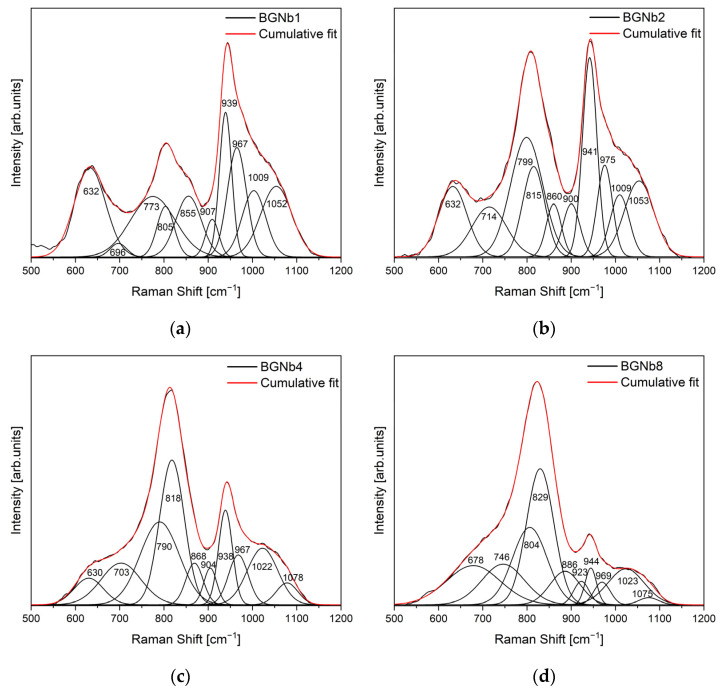
Deconvoluted Raman spectra of (**a**) BGNb1; (**b**) BGNb2; (**c**) BGNb4; and (**d**) BGNb8.

**Figure 5 ijms-24-05244-f005:**
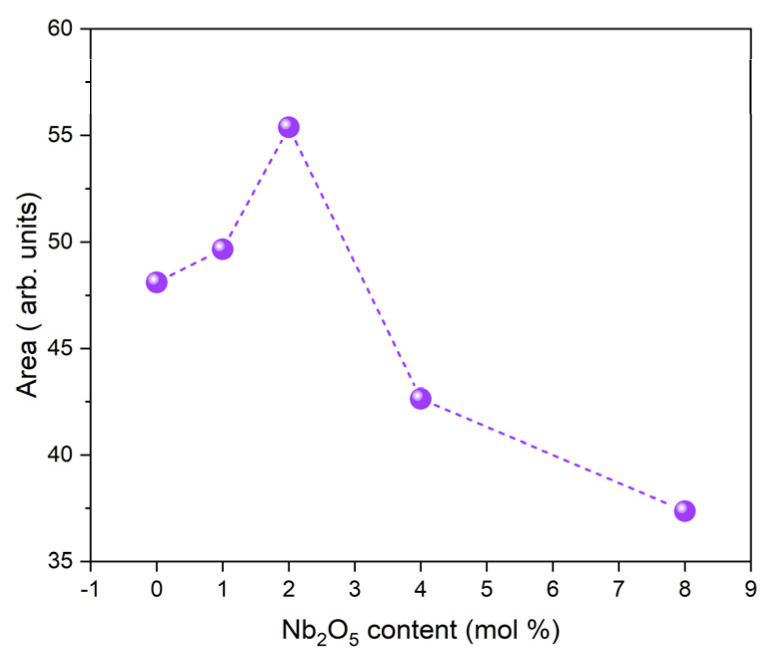
Sum of the areas of the bands associated with NBOs vibrations.

**Figure 6 ijms-24-05244-f006:**
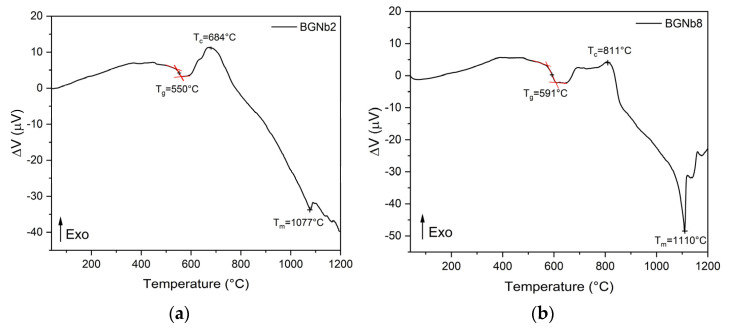
DTA spectra of (**a**) BGNb2; (**b**) BGNb8.

**Figure 7 ijms-24-05244-f007:**
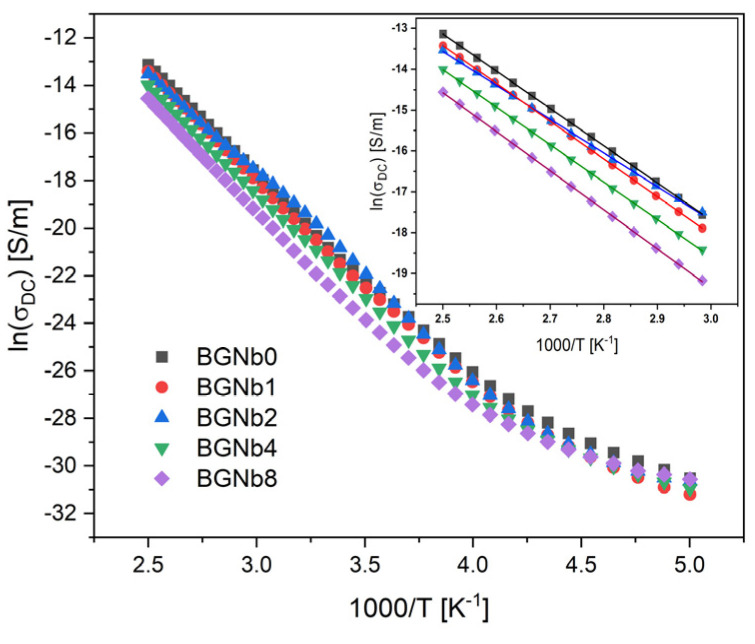
DC conductivity versus 1000/T (inset: a magnification of the high-temperature measurement zone is shown in the inset; the lines reflect the Arrhenius fit).

**Figure 8 ijms-24-05244-f008:**
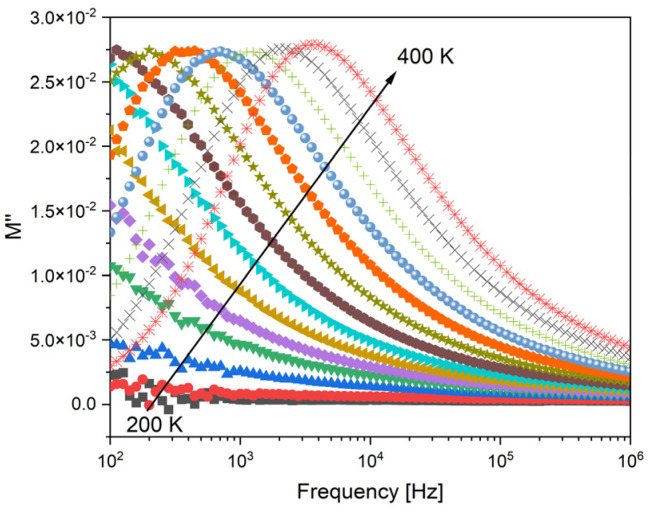
The imaginary part of the dielectric modulus (M″) versus frequency for BGNb2 glass, between 200 K and 400 K.

**Figure 9 ijms-24-05244-f009:**
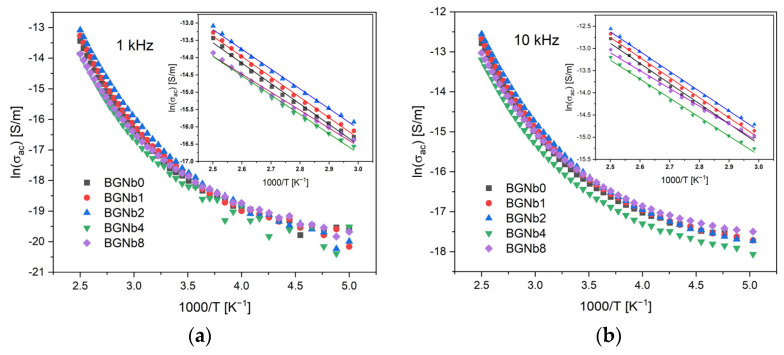
AC conductivity versus 1000/T at (**a**) 1 kHz; (**b**) 10 kHz (inset: magnification of the high-temperature measurement zone; the lines reflect the Arrhenius fit).

**Figure 10 ijms-24-05244-f010:**
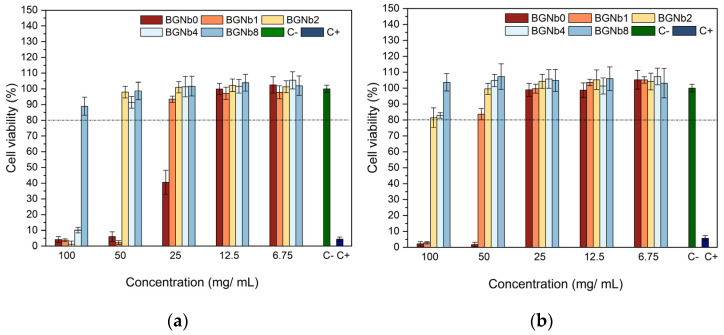
Relative viability of (**a**) non-passivated and (**b**) passivated bioactive glass samples modified by Nb_2_O_5_ insertion, in culture with Saos-2 cells.

**Figure 11 ijms-24-05244-f011:**
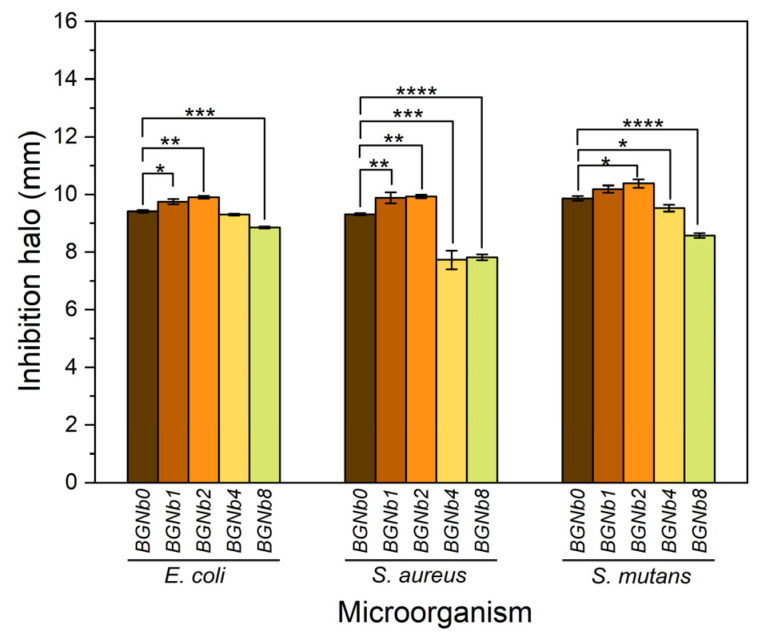
Antibacterial activity of the bioglass samples against *E. coli*, *S. aureus,* and *S. mutans* based on inhibition halo. Results are reported as mean ± SD. The asterisks indicate significance in an unpaired *t*-test; * *p* ≤ 0.05; ** *p* ≤ 0.01; *** *p* ≤ 0.001; **** *p* ≤ 0.0001.

**Figure 12 ijms-24-05244-f012:**
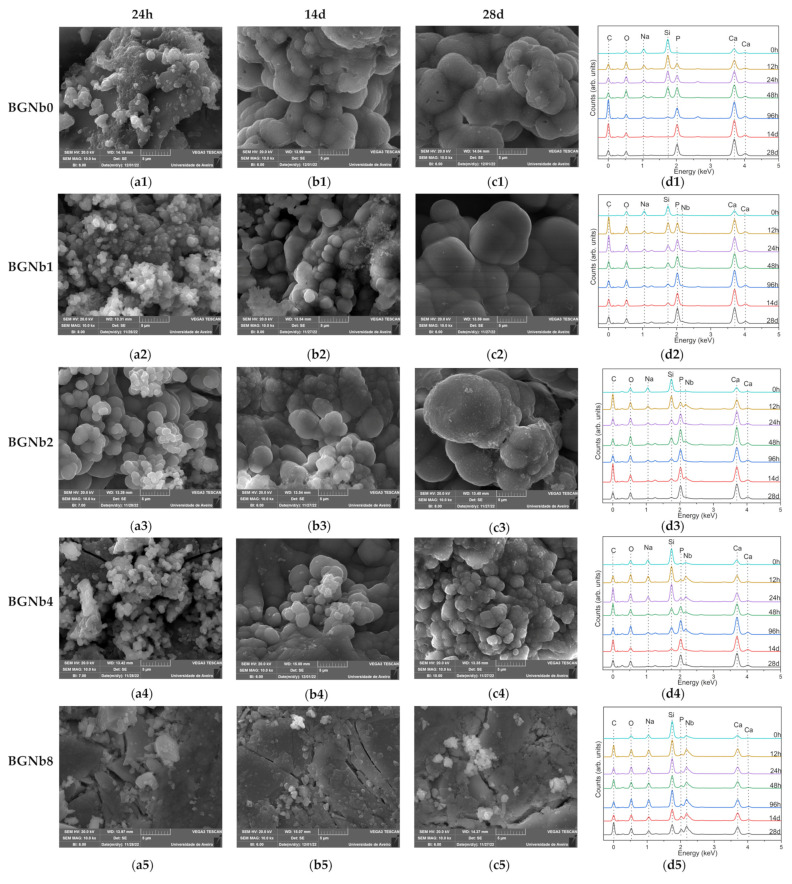
SEM micrographs of five groups of bioactive glass samples modified by the insertion of different percentages of Nb_2_O_5_ after immersion in SBF for (**a1**–**a5**) 24 h; (**b1**–**b5**) 14 d; (**c1**–**c5**) 28 d; (**d1**–**d5**) plots of the EDS analysis for all immersion times. (The magnification of SEM images is 10 kX).

**Figure 13 ijms-24-05244-f013:**
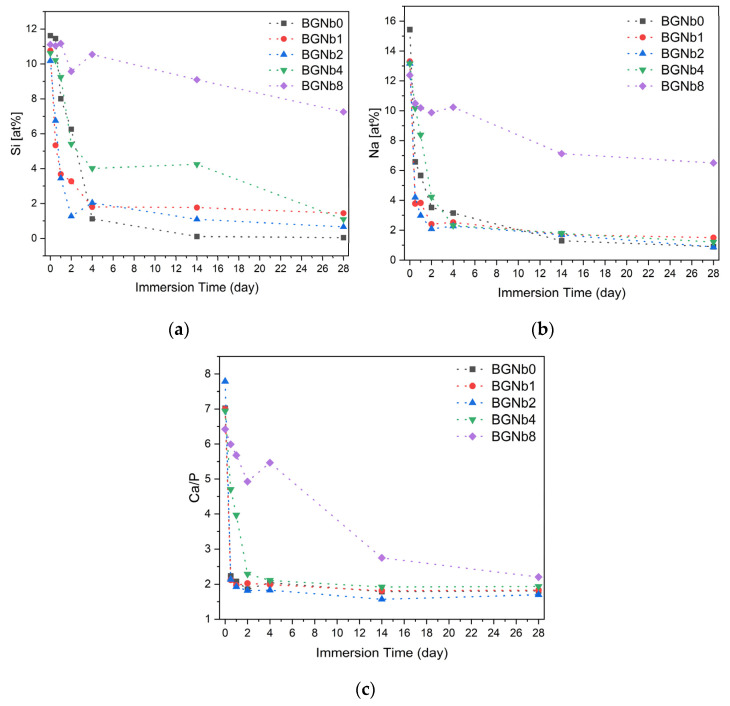
The variation of elemental concentration of (**a**) Si; (**b**) Na; (**c**) Ca/P ratio on the surface of bioactive glasses after immersion in SBF as a function of immersion time.

**Figure 14 ijms-24-05244-f014:**
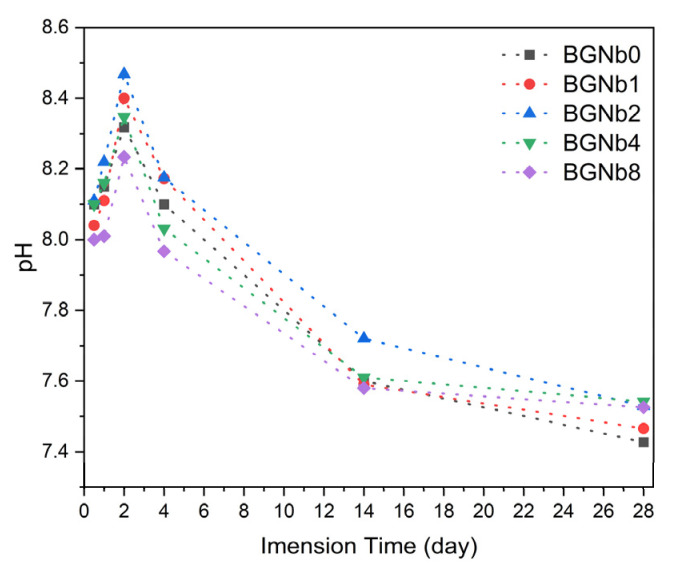
Variation of pH of the SBF solution with the immersion time.

**Table 1 ijms-24-05244-t001:** The characteristic temperatures for BGNb0, BGNb2, and BGNb8.

Sample	T_g_ (°C)	T_c_ (°C)	T_m_ (°C)
BGNb0 [3]	552	728	1175
BGNb2	550	684	1077
BGNb8	591	811	1110

**Table 2 ijms-24-05244-t002:** The DC conductivity (σ_DC_), DC activation energy E_a_ (DC), AC conductivity (σ_AC_), and AC activation energy E_a_ (AC), for all bioglass samples.

Sample	σ_DC_ (×10^−9^) [S/m] (At 300 K)	E_a_ (DC) [kJ/mol]	σ_AC_ (×10^−7^) [S/m] (At 350 K, 1 kHz)	E_a_ (AC) [kJ/mol] (At 1 kHz)	σ_AC_ (×10^−7^) [S/m] (At 350 K, 10 kHz)	E_a_ (AC) [kJ/mol] (At 10 kHz)
BGNb0	0.91 ± 0.01	75.42 ± 0.08	1.50 ± 0.01	49.39 ± 0.98	4.91 ± 0.05	37.95 ± 0.97
BGNb1	0.78 ± 0.04	76.79 ± 0.06	1.84 ± 0.04	50.21 ± 0.96	5.65 ± 0.11	38.49 ± 0.79
BGNb2	1.52 ± 0.10	69.35 ± 0.20	2.36 ± 0.03	48.61 ± 0.87	6.54 ± 0.13	37.83 ± 0.61
BGNb4	0.51 ± 0.02	76.47 ± 0.15	1.17 ± 0.01	47.55 ± 0.86	3.69 ± 0.06	36.37 ± 0.66
BGNb8	0.19 ± 0.01	79.47 ± 0.16	1.23 ± 0.02	44.25 ± 0.92	4.60 ± 0.21	34.11 ± 0.69

## Data Availability

The data presented in this study are available on request from the corresponding author.

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
