# Peer review of "Extensive Investigation on the Effect of Niobium Insertion on the Physical and Biological Properties of 45S5 Bioactive Glass for Dental Implant"

_ijms, 2023, doi:10.3390/ijms24065244_

Round 1

Reviewer 1 Report

The writing of the paper is very good, the authors made an excellent  presentation and appropriate discussion of the results, as the references present up-to-date information.

Author Response

The authors thank the reviewer for the comments/appreciation of the manuscript, "The writing of the paper is very good, the authors made an excellent  presentation and appropriate discussion of the results, as the references present up-to-date information."

While there was a suggestion to improve the language in the manuscript, the authors made an effort to reduce the errors and highlighted them as suggested by the other reviewers also. The authors also would appreciate the specific suggestions if any by the reviewer to improve the quality of the manuscript. 

Reviewer 2 Report

Dear authors, I wish to appreciate you for your work. The work is interesting. But some technical issues need to be addressed:

- The Author stated in the Result and discussion: "The XRD patterns illustrated in Figure 1, consist of broad bands which confirm the amorphous character of the glasses. But other peak? Please, give more information about them.

- In vitro cytotoxicity testing is an indispensable part of the development of new biomaterials. However, the standard ISO 10993-5 enables variability in the testing conditions, which makes the results of the test incomparable. Therefore, when performing in vitro cytotoxicity testing of biomaterials, the authors should carefully specify the conditions of the test and comparison of different studies should be carried out with caution. The preparation of extracts and other conditions of the test are not strictly defined. The Authors must include more information (for example summarized in Table).

The paper is well organized and the results is of great fundamental importance. I recommend acceptance for publication in the International Journal of Molecular Sciences after minor revision.

Reviewer 3 Report

The article deals with the effectiveness of the Nb contained in a bioglass (that of Hench the 45S5) on the biological properties to be used as a bone implant in dental surgery.

For this, various tests (biocompatibility, bacterial activity and bioactivity) and physico-chemical characterizations (Tg, electrical conductivity, structure by infrared and Raman) were carried out in order to observe on the one hand the influence of the rate of Nb on these properties, and on the other to establish and understand the evolution of biological properties with these physico-chemical properties. It appears that the composition with 2% Nb2O5 is the most promising for applications as a dental implant.

A few remarks, questions

1 – What is the uncertainty in determining the glass transition temperature?

2- Show an example of an impedance diagram for the measurements of electrical properties in alternating current, why measured only at 1 kHz?

3- Figure 9: define in the legend C+ and C-

4- We observe in figure 11, on the EDS spectra the Nb peak, do we have an idea in which chemical form the Nb appears in the apatite layer?

5- P2O5 was used as a reagent and source of phosphorus, what precautions were taken because it is a hygroscopic product? would NaPO3 have been better since P2O5 is also volatile? have chemical analyzes been done to verify at least one chemical composition?

6- Bioactivity tests: tests made on powder, has the powder been sieved, a particle size class has been chosen?

7- Why does the materials and methods part come after the discussion? Not logical
